# Association between restless legs syndrome and sleep quality in Peruvian medical students

**Rubí Paredes-Angeles[1], Cesar Copaja-Corzo**  [2,3*], **Alvaro Taype-Rondan[1,2]**

**1** EviSalud – Evidencias en Salud, EsSalud, Lima, Peru, **2** Unidad de Investigación para la Generación y Síntesis de Evidencias en Salud, Universidad San Ignacio de Loyola, Lima, Peru, **3** Servicio de Infectología, Hospital Nacional Edgardo Rebagliati Martins, EsSalud, Lima, Peru

\* csarcopaja@gmail.com

## Abstract

### Objective

To evaluate the association between restless legs syndrome (RLS) and sleep quality in Peruvian medical students.

### Methods

Cross sectional study with a secondary data analysis. The study included Peruvian medical students surveyed in 2020. The outcome was sleep quality evaluated using the Pittsburgh Sleep Quality Index (PSQI), and the exposure variable was RLS assessed with the International Restless Legs Syndrome Study Group (IRLSSG) scale. To address the research question, we employed Poisson regression models with robust variance.

### Results

We analyzed information from 3139 medical students (61.1% female, median age 22.3 years). 15.3% experienced symptoms of Restless Legs Syndrome (RLS), and 77.2% had poor sleep quality. The prevalence of poor sleep quality was higher in female participants (80.0%) and those with symptoms of anxiety (92.8%), depression (91.6%), and severe nomophobia (86.3%). In the multivariable model, the presence of RLS symptoms was associated with poor sleep quality (prevalence ratio: 1.05, 95% CI 1.01 - 1.09, p < 0.013).

### Conclusion

We found a high prevalence of poor sleep quality, notably associated with RLS. Other factors associated with poor sleep quality were the academic year of study, anxiety, depression, and nomophobia.

**Data availability statement:** All relevant data are within the paper and its Supporting Information files.

**Funding:** The Universidad San Ignacio de Loyola finances the article processing charge. Funding Acquisition was by authors C.C.C. The funders had no role in study design, data collection and analysis, decision to publish, or preparation of the manuscript.

**Competing interests:** The authors have declared that no competing interests exist.

## Introduction

Restless legs syndrome (RLS) is a chronic neurological condition characterized by an irresistible need to move the legs to relieve unpleasant sensations such as tingling, prickling, or burning sensations [1]. The International Restless Legs Syndrome Study Group has proposed four criteria for the diagnosis of RLS: frequent leg movements with abnormal sensitivity in the skin of the legs, temporary relief of unpleasant symptoms when moving the legs, onset or worsening of symptoms with rest or not moving the legs, and onset or worsening of symptoms at night [2].

The prevalence of RLS in South America has been reported at approximated 16%, surpassing rates observed in other regions [2]. Furthermore, it is noteworthy that RLS may disproportionately impact individuals experiencing elevated stress levels, a demographic that includes medical students [3].

It has been proposed that RLS would affect sleep quality by causing periodic leg movements before and during sleep [4]. Poor quality of sleep is manifested through sleep interruptions, difficulties in falling asleep, frequent awakenings during the night, and excessive daytime sleepiness [5]. This condition affects the quality of life as it can trigger a variety of physical and psychological problems such as anxiety, depression, fatigue, memory, and concentration problems [6]. It is estimated that the global prevalence of inadequate sleep quality is 52.7% among medical students, with Peru reporting a prevalence of up to 83.9% for inadequate sleep quality [7], which appears to be higher compared to other groups of university students and the general population [8]. This could be due to several particular challenges faced by medical students, such as academic overload, the need to meet clinical practice requirements, exposure to high-stress situations, and lack of time for personal activities [9].

Periodic leg movements can occur during sleep or relaxed wakefulness, the latter in more severe forms when Restless Legs Syndrome (RLS) is present [8]. An association between RLS and sleep disturbances is suspected due to the increase in these leg movements during sleep, but the evidence supporting this hypothesis is scarce [10] and heterogeneous, particularly among medical students, [11,12] with small studies [13] and a lack of control for potential confounding variables [11,13]. Given the high prevalence of poor sleep quality among medical students, evaluating this association is necessary to gather information with which preventive and therapeutic interventions can be developed. Thus, the aim of this study was to evaluate the association between Restless Legs Syndrome and sleep quality in Peruvian medical students.

## Methods

### Study design

This cross-sectional study conducted a secondary analysis of data derived from the "Nomophobia and its associated factors in Peruvian medical students" project. The primary study was a cross-sectional analytical study aimed at identifying the factors associated with nomophobia among medical students in Peru. It was conducted in 38 medical schools and surveyed 3,139 students [14].

### Participants and procedures

In Peru, the human medicine degree studies extend over seven years, with the final year dedicated to a practical rotation is known as internship. In 2020, there were an estimated 30,000 medical students in Peru, spread across 46 universities throughout the country [15].

This study targeted adult medical students (aged 18 and above) in Peru who voluntarily participated and confirmed their enrollment in a human medicine career. Exclusion criteria

encompassed students who did not possess a mobile phone in the month in which the survey was performed.

Convenience sampling was employed in the primary study, utilizing an online survey conducted through Google Forms. The primary study collected data in two stages between June 2020 and March 2021. This was done virtually through social media platforms (Facebook, WhatsApp) because universities in Peru held online classes during the pandemic to minimize the risk of contagion.

The data collection occurred in two stages. In the first stage, a Facebook page was created to promote the research project. An open invitation was extended to all medical students in Peru through posters distributed in private Facebook groups for medical students, as well as through paid Facebook ads specifically targeting medical students in Peru. In the second stage, 23 students from various medical schools were recruited through networks within the Scientific Society of Medical Students of Peru. The recruited students attended a 45-minute virtual meeting, where they were trained to contact students from their respective universities.

In order for the students to participate in the survey, they first had to provide their informed consent. As an incentive, all students who completed the survey were given free access to a Google Drive folder with a collection of resources from medical courses. The survey is available in S1 File.

To develop our study, we accessed the database in April 2023 and conducted the analyses and reporting of results on the same date.

## Variables and measures

Our outcome (dependent variable) of interest was sleep quality. This variable was measured with the Pittsburgh Sleep Quality Index (PSQI) [16]. The PSQI is the most widely used measure for assessing sleep quality, useful for epidemiological studies and easy to complete [17]. A weakness is that different factorial structures have been found in the samples where it has been applied [17]. The 19 PSQI items are grouped into 7 components: sleep duration, sleep disturbance, sleep latency, daytime dysfunction due to sleepiness, sleep efficiency, overall sleep quality, and sleep medication use. Each component is evaluated using a scale ranging from 0 to and the overall rating is derived from the cumulative scores of the components, with an overall score ranging from 0 to 21, and a higher score reflecting poorer sleep quality. Individuals scoring from 0 to 5 points were deemed to have adequate sleep quality, while those scoring 6 to 21 points were classified as poor sleep quality [18]. The study utilized the Peruvian-validated version, which demonstrated optimal internal consistency, as evidenced by a Cronbach's alpha of 0.81 [19]. In our sample, the Cronbach's alpha was 0.75.

The RLS was the exposure of interest (independent variable). To measure RLS symptoms, the scale based on the criteria of the International Restless Legs Syndrome Study Group (IRLSSG) was used. This scale consists of 3 items with scores ranging from 0 to 3 points. The items ask: 1. Do you have a feeling of discomfort or discomfort in your legs, with the urgency or need to move them? 2. Do these discomforts occur only at rest and improve with movement? 3. Are they worse in the evening than in the morning? The variable was categorized according to the following criteria: no presence of symptoms of RLS (< 3 points) and presence of RLS (= 3 points), since a positive response to the 3 items indicates the presence of RLS [20]. This measure has a sensitivity of 100% and specificity of 88% in a Spanish version [21].

Confounding variables. Additional variables considered for the analysis included: age (in tertiles), sex (male or female), university financing (public or private), year of study (first, second, third, fourth or fifth), university location (cities according to urbanity), engagement in physical activity (yes or no), practicing a sport (yes or no), anxiety symptoms (yes or

no), depression symptoms (yes or no) and nomophobia symptoms (no nomophobia, mild nomophobia, moderate nomophobia or severe nomophobia).

To assess anxiety and depression symptoms, the Hopkins Symptom Checklist-25 (HSCL-25) was employed [22]. HSCL-25 is widely used for both clinical and epidemiological, self-administered, easy-to-use for participants and researchers and simple of interpretation [23]. It does not provide a clinical diagnosis; it only assesses early symptoms of anxiety and depression [23]. This checklist comprises 25 items, with 10 focusing on anxiety and 15 on depression. Responses are rated on a 4-point Likert scale, ranging from 1 ("not at all") to 4 ("a lot"). If participants has an average score above 1.75 on the anxiety or depression subscales of the HSCL-25, they are considered to have possible symptoms of anxiety or depression, respectively [24]. The HSCL has been validated in Spanish in an adult Peruvian population, demonstrating satisfactory internal consistency (anxiety, Cronbach's alpha = 0.81; and depression, Cronbach's alpha = 0.86) [24]. This study used the version of the Peruvian Ministry of Health [25]. For our sample, Cronbach's alpha was 0.90 for anxiety and 0.93 for depression.

Nomophobia symptoms were measured using the Nomophobia Questionnaire (NMP-Q) [26]. The NMP-Q comprises 20 items and four factors: "not being able to access information", "giving up convenience", "not being able to communicate" and "loss of connection". The questionnaire uses a Likert scale with a scoring range of 7 points, from 1 ("strongly disagree") to 7 ("strongly agree"), resulting in a total score falling between 20 and 140 points. The following cut-off points were applied for interpreting the NMP-Q questionnaire: absence (a score of 20), mild (greater than 20 but less than 60), moderate (a score of 60 or higher but less than 100) and severe (a score of 100 or higher). A systematic review found that the NMP-Q demonstrates excellent internal consistency, with strong structural validity across its four-factor model [27]. In this study, the Spanish version of the NMP-Q was employed, having undergone validation and exhibiting satisfactory internal consistency (Cronbach's alpha values ranged from 0.78 to 0.92) [28]. In our sample, Cronbach's alpha was 0.88 for the factor "not being able to access information", 0.87 for "giving up convenience", 0.95 for "not being able to communicate", and 0.93 for "loss of connection".

## Data analysis

We performed the statistical analysis using the Stata v17 software. First, for the descriptive analysis, frequencies and percentages were employed. Second, in the bivariate analysis, the sleep quality was associated with the rest of the variables, using the Chi-square statistical test. Third, for the multivariable analysis, Poisson regression models with robust variance were used to calculate prevalence ratios (PR) with their respective 95% confidence intervals (95% CI). We used this regression model since the frequency of inadequate sleep quality was high so using logistic regression (i.e., calculating odds ratios) could overestimate the objective association [29]. It is essential to emphasize that only the estimate of the association between RLS and sleep quality should be interpreted, while the estimates for the confounding variables should not be considered [30]. Those variables exhibiting a statistically significant association with the outcome (p < 0.05) were subsequently included in the adjusted regression model.

## Ethics

This study received ethical approval from the Universidad Peruana de Tacna, Peru (Approval record: 99/FACSA/UI). Participation was voluntary and all the informants provided an informed consent. The procedures used in this study adhere to the tenets of the Declaration of Helsinki.

## Results

Data of 3139 medical students were analyzed. Most of the participants were women (61.1%) and the median age was 22.3 years (25th percentile - 75th percentile: 20 - 24 years). Around half of them (50.8%) studied at a private university, and around one-third (29.9%) were in their fifth year of studies. Anxiety and depression symptoms were identified in 34.8% and 42.8% respectively. Additionally, 15.3% presented RLS symptoms, while 77.2% had poor sleep quality (Table 1).

A higher prevalence of poor sleep quality was found in female participants (80.0%) and those who presented symptoms of anxiety (92.8%), depression (91.6%), and severe nomophobia (86.3%) (p < 0.001). On the other hand, a lower prevalence of poor sleep quality was found in those who did physical activity (72.4%) and practiced an sport (73.3%) (p < 0.001). There was a higher prevalence of inadequate sleep quality in those who had RLS symptoms (87.9%) compared to those who did not have RLS symptoms (75.3%), and this difference was significant (p < 0.001) (Table 2).

In the crude regression analysis, restless legs syndrome was associated with sleep quality (PR: 1.17, 95% CI 1.12 - 1.21, p < 0.001). After adjusting for sex, age, type of university, year of study, university location, doing physical activity, practicing a sport, anxiety, depression and nomophobia symptoms, the association between RLS and sleep quality remained significant (PR: 1.05, 95% CI 1.01 - 1.09, p < 0.013) (Table 3).

## Discussion

This is the first study to evaluate the relationship between restless legs syndrome and sleep quality among medical students, with a larger sample size than that used in previous research. Findings showed that students with RLS symptoms had a 5% higher prevalence of poor sleep quality than those without RLS. Moreover, the prevalence of poor sleep quality was very high (77.2%).

Regarding the high prevalence of poor sleep quality, previous studies have found similar results. Two systematic reviews, which conducted their searches in 2018, reported prevalences of 55.0% [8] and 52.7% [31] for poor sleep. Although it did not include Peruvian studies, one of them found that America was the second region with the highest prevalence (59.9%) [31]. The high prevalence reported in our study may be due to high demands of the medical education, lack of learning strategies, or certain behaviors such as procrastination, which translate into few hours of sleep [32].

Furthermore, it is necessary to highlight that our data was collected during the COVID-19 pandemic, which exacerbated the sleep problems of students, including insomnia, early awakening, and difficulty falling asleep [33]. In this context, individuals, such as medical students, were forced to change their lifestyles and adapt to virtual environments, which were rarely used before the pandemic in Peruvian universities [34]. In Peru, medical students in their final years of study typically undergo their academic training primarily in hospitals; however, due to the COVID-19 pandemic, access to hospitals was restricted. Additionally, to curb the spread of the virus, access to universities was limited. As a result, all classes transitioned to virtual formats, a practice uncommon in Peruvian universities before the pandemic [34]. This situation could have decreased the quality of sleep and increased insomnia among medical students, potentially impacting not only their academic performance but also their overall health [35].

We found that those who had RLS had worse sleep quality. Only four studies have evaluated this association in medical students, two of which were conducted in Türkiye. The first one, carried out in a hospital with sixth-year students and resident physicians, developed a

**Table 1. Characteristics of the participants (n = 3139).**

| Characteristic | n (%) |
|---|---|
| Sex | |
| Male | 1220 (38.9) |
| Female | 1919 (61.1) |
| Age in years* | 22.3 (20.0 - 24.0) |
| 18 to 20 | 1086 (34.6) |
| 21 to 23 | 1162 (37.0) |
| 24 to more | 891 (28.4) |
| Type of university | |
| Public | 1544 (49.2) |
| Private | 1595 (50.8) |
| University location | |
| Lima | 746 (23.8) |
| Arequipa, Trujillo, Chiclayo | 615 (19.6) |
| Piura, Huancayo, Cusco | 813 (25.9) |
| Other | 965 (30.7) |
| Year of study | |
| First | 300 (9.6) |
| Second | 670 (21.3) |
| Third | 639 (20.4) |
| Fourth | 593 (18.9) |
| Fifth | 937 (29.8) |
| Doing physical activity | |
| No | 1962 (62.5) |
| Yes | 1177 (37.5) |
| Practicing an sport | |
| No | 1660 (52.9) |
| Yes | 1479 (47.1) |
| Anxiety symptoms (HSCL-25) | |
| No | 2048 (65.2) |
| Yes | 1091 (34.8) |
| Depression symptoms (HSCL-25) | |
| No | 1796 (57.2) |
| Yes | 1343 (42.8) |
| Nomophobia (NMP-Q) | |
| No nomophobia | 125 (4.0) |
| Mild nomophobia | 1974 (63.0) |
| Moderate nomophobia | 807 (25.7) |
| Severe nomophobia | 233 (7.3) |
| Restless legs syndrome | |
| No | 2660 (84.7) |
| Yes | 479 (15.3) |
| Sleep quality (PSQI) | |
| Good | 714 (22.8) |
| Poor | 2425 (77.2) |

HSCL-25: Hopkins Symptom Checklist-25. NMP-Q: Nomophobia Questionnaire.

*Median (25th percentile - 75th percentile).

**Table 2. Characteristics associated with sleep quality (n = 3139).**

| Characteristic | Sleep quality | | p* |
|---|---|---|---|
| | Good (n = 714) | Poor (n = 2425) | |
| | n (%) | n (%) | |
| Sex | | | <0.001 |
| Male | 331 (27.1) | 889 (72.9) | |
| Female | 383 (20.0) | 1536 (80.0) | |
| Age | | | <0.001 |
| 18 to 20 | 211 (19.4) | 875 (80.6) | |
| 21 to 23 | 259 (22.3) | 903 (77.7) | |
| 24 to more | 244 (27.4) | 647 (72.6) | |
| University type | | | 0.005 |
| Public | 384 (24.9) | 1160 (75.1) | |
| Private | 330 (20.7) | 1265 (79.3) | |
| University location | | | 0.241 |
| Lima | 165 (22.1) | 581 (77.9) | |
| Arequipa, Trujillo, Chiclayo | 126 (20.5) | 489 (79.5) | |
| Piura, Huancayo, Cusco | 184 (22.6) | 629 (77.4) | |
| Other | 239 (24.8) | 726 (75.2) | |
| Year of study | | | 0.001 |
| First | 56 (18.7) | 244 (81.3) | |
| Second | 137 (20.5) | 533 (79.5) | |
| Third | 128 (20.0) | 511 (80.0) | |
| Fourth | 138 (23.3) | 455 (76.7) | |
| Fifth | 255 (27.2) | 682 (72.8) | |
| Doing physical activity | | | <0.001 |
| No | 389 (19.8) | 1573 (80.2) | |
| Yes | 325 (27.6) | 852 (72.4) | |
| Practicing an sport | | | <0.001 |
| No | 319 (19.2) | 1341 (80.8) | |
| Yes | 395 (26.7) | 1084 (73.3) | |
| Anxiety symptoms (HSCL-25) | | | <0.001 |
| No | 627 (30.6) | 1421 (69.4) | |
| Yes | 87 (8.0) | 1004 (92.8) | |
| Depression symptoms (HSCL-25) | | | <0.001 |
| No | 601 (33.5) | 1195 (66.5) | |
| Yes | 113 (8.4) | 1230 (91.6) | |
| Nomophobia (NMP-Q) | | | <0.001 |
| No nomophobia | 84 (67.2) | 41 (32.8) | |
| Mild nomophobia | 474 (24.0) | 1500 (76.0) | |
| Moderate nomophobia | 124 (15.4) | 683 (84.6) | |
| Severe nomophobia | 32 (13.7) | 201 (86.3) | |
| Restless legs syndrome | | | <0.001 |
| No | 656 (24.7) | 2004 (75.3) | |
| Yes | 58 (12.1) | 421 (87.9) | |

HSCL-25: Hopkins Symptom Checklist-25. NMP-Q: Nomophobia Questionnaire

*Chi-square, 95% confidence interval

**Table 3. Association between restless legs syndrome and sleep quality in Peruvian medical students (n = 3139).**

| Characteristic | Crude PR | 95% CI | p | Adjusted PR | 95% CI | p |
|---|---|---|---|---|---|---|
| Sex | | | | | | |
| Male | Ref. | | | Ref. | | |
| Female | 1.10 | 1.05 - 1.14 | <0.001 | 1.04 | 1.00 - 1.08 | 0.067 |
| Age | | | | | | |
| 18 to 20 | Ref. | | | Ref. | | |
| 21 to 23 | 0.96 | 0.92 - 1.01 | 0.095 | 1.01 | 0.96 - 1.06 | 0.744 |
| 24 to more | 0.90 | 0.86 - 0.95 | <0.001 | 1.01 | 0.96 - 1.07 | 0.683 |
| University type | | | | | | |
| Public | Ref. | | | Ref. | | |
| Private | 1.06 | 1.02 - 1.10 | 0.005 | 1.02 | 0.99 - 1.06 | 0.234 |
| University location | | | | | | |
| Lima | Ref. | | | — | — | — |
| Arequipa, Trujillo, Chiclayo | 1.02 | 0.97 - 1.08 | 0.464 | — | — | — |
| Piura, Huancayo, Cusco | 0.99 | 0.94 - 1.05 | 0.808 | — | — | — |
| Other | 0.97 | 0.92 - 1.02 | 0.198 | — | — | — |
| Year of study | | | | | | |
| First | Ref. | | | Ref. | | |
| Second | 0.98 | 0.92 - 1.05 | 0.514 | 0.95 | 0.90 - 1.02 | 0.191 |
| Third | 0.98 | 0.92 - 1.05 | 0.619 | 0.97 | 0.90 - 1.03 | 0.317 |
| Fourth | 0.94 | 0.88 - 1.01 | 0.103 | 0.96 | 0.89 - 1.03 | 0.255 |
| Fifth | 0.89 | 0.84 - 0.96 | 0.001 | 0.91 | 0.84 - 0.98 | 0.013 |
| Doing physical activity | | | | | | |
| No | Ref. | | | Ref. | | |
| Yes | 0.90 | 0.87 - 0.94 | <0.001 | 0.95 | 0.91 - 0.99 | 0.020 |
| Practicing an sport | | | | | | |
| No | Ref. | | | Ref. | | |
| Yes | 0.91 | 0.87 - 0.94 | <0.001 | 0.99 | 0.95 - 1.03 | 0.549 |
| Anxiety symptoms (HSCL-25) | | | | | | |
| No | Ref. | | | Ref. | | |
| Yes | 1.33 | 1.28 - 1.37 | <0.001 | 1.11 | 1.06 - 1.15 | <0.001 |
| Depression symptoms (HSCL-25) | | | | | | |
| No | Ref. | | | Ref. | | |
| Yes | 1.38 | 1.33 - 1.43 | <0.001 | 1.22 | 1.17 - 1.28 | <0.001 |
| Nomophobia (NMP-Q) | | | | | | |
| No nomophobia | Ref. | | | Ref. | | |
| Mild nomophobia | 2.32 | 1.80 - 2.98 | <0.001 | 2.10 | 1.64 - 2.69 | <0.001 |
| Moderate nomophobia | 2.58 | 2.00 - 3.32 | <0.001 | 2.19 | 1.71 - 2.81 | <0.001 |
| Severe nomophobia | 2.63 | 2.04 - 3.40 | <0.001 | 2.20 | 1.70 - 2.83 | <0.001 |
| Restless legs syndrome | | | | | | |
| No | Ref. | | | Ref. | | |
| Yes | 1.17 | 1.12 - 1.21 | <0.001 | 1.05** | 1.01 - 1.09 | 0.013 |

HSCL-25: Hopkins Symptom Checklist-25. NMP-Q: Nomophobia Questionnaire.

PR: Prevalence ratio. 95% CI: 95% confidence interval. —: Variable not included in the multivariable model because it was not considered in the adjustment.

*Adjusted for sex, age, year of study, doing physical activity, practicing a sport, anxiety symptoms, depression symptoms and nomophobia.

**Only this adjusted RP should be interpreted, the rest of adjusted RP correspond to the confounding variables so they should not be interpreted.

multivariable model in which no association was found between both variables [12], while in the second study, also conducted in Turkey, only a simple bivariate analysis was performed, and a significant association was found [35]. It is important to highlight that the sample sizes in these studies were 341 and 402, respectively, [12,35] while in our study 3139 medical students were included.

The third study was conducted on health sciences students (n = 148) in Saudi Arabia, of which almost 50% were medical students [13]. A significant correlation was found between RLS symptoms and sleep quality; however, multivariable analyzes were not performed [13]. Finally, the fourth study was conducted on 256 health students in Peru, including medical students, and reported that the odds of sleeping poorly in those who had restless legs syndrome were 5 times higher than in those who did not have the syndrome, but the association was not statistically significant [11]. Although an association was found in our study, the difference between those who present RLS and those who do not is very small (5%). Therefore, more research may be necessary for more conclusive results, especially in this population. In other populations, such as pregnant women [20], this association has been studied with robust models, and significant results have been found.

The limitations of this study should be considered. First, a non-probabilistic convenience sampling was performed, so the subject selection may not be representative of the population. Second, self-report measures were used for assessing RLS, sleep quality, and some covariates (depression, anxiety, and nomophobia). In response to this, validated and adapted measures were used. To address the potential social desirability bias that could arise from the use of these instruments, participants were informed that the survey was anonymous, and their information was confidential. Third, this is a cross-sectional study, so the temporal cause-effect relationship of the variables of interest cannot be verified. Fourth, other variables that could influence in the association of interest, such as stress and the use of psychoactive substance, were not included.

In conclusion, our study performed in Peruvian medical students revealed a prevalent issue of poor sleep quality, notably associated to RLS. Additionally, factors such as the academic year of study, anxiety, depression, and nomophobia were identified as variables linked to this diminished sleep quality.

## Supporting information

**S1 File.  Study survey.**
(DOCX)

**S2 File.  Study database.**
(XLSX)

## Author contributions

**Conceptualization:** Rubí Paredes-Angeles, Cesar Copaja-Corzo.

**Data curation:** Rubí Paredes-Angeles.

**Formal analysis:** Rubí Paredes-Angeles, Cesar Copaja-Corzo, Alvaro Taype-Rondan.

**Investigation:** Alvaro Taype-Rondan.

**Methodology:** Cesar Copaja-Corzo, Alvaro Taype-Rondan.

**Supervision:** Alvaro Taype-Rondan.

**Validation:** Rubí Paredes-Angeles.

**Writing – original draft:** Rubí Paredes-Angeles, Cesar Copaja-Corzo.

**Writing – review & editing:** Rubí Paredes-Angeles, Cesar Copaja-Corzo, Alvaro Taype-Rondan.

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
