## [Decision Letter · Decision Letter 0]

12 Aug 2024

PONE-D-24-08629Association between restless legs syndrome and sleep quality in Peruvian medical studentsPLOS ONE

Dear Dr. Copaja-Corzo,

Thank you for submitting your manuscript to PLOS ONE. After careful consideration, we feel that it has merit but does not fully meet PLOS ONE’s publication criteria as it currently stands. Therefore, we invite you to submit a revised version of the manuscript that addresses the points raised during the review process.

We look forward to receiving your revised manuscript.

Kind regards,

Runtang Meng, PhD

Academic Editor

PLOS ONE

2. Please ensure that you have specified a) Did participants provide their written or verbal informed consent to participate in this study?

Reviewers' comments:

Reviewer's Responses to Questions

**Comments to the Author**

1. Is the manuscript technically sound, and do the data support the conclusions?

Reviewer #1: Yes

Reviewer #2: Yes

Reviewer #3: Yes

Reviewer #4: Yes

Reviewer #5: Partly

Reviewer #6: Yes

Reviewer #7: Yes

Reviewer #8: Partly

Reviewer #9: Yes

Reviewer #10: Partly

2. Has the statistical analysis been performed appropriately and rigorously? 

Reviewer #1: Yes

Reviewer #2: I Don't Know

Reviewer #3: Yes

Reviewer #4: Yes

Reviewer #5: No

Reviewer #6: Yes

Reviewer #7: Yes

Reviewer #8: Yes

Reviewer #9: Yes

Reviewer #10: No

3. Have the authors made all data underlying the findings in their manuscript fully available?

Reviewer #1: Yes

Reviewer #2: Yes

Reviewer #3: Yes

Reviewer #4: Yes

Reviewer #5: Yes

Reviewer #6: No

Reviewer #7: Yes

Reviewer #8: Yes

Reviewer #9: Yes

Reviewer #10: Yes

4. Is the manuscript presented in an intelligible fashion and written in standard English?

Reviewer #1: Yes

Reviewer #2: Yes

Reviewer #3: Yes

Reviewer #4: Yes

Reviewer #5: Yes

Reviewer #6: Yes

Reviewer #7: Yes

Reviewer #8: Yes

Reviewer #9: Yes

Reviewer #10: No

5. Review Comments to the Author

Reviewer #1: Overall, this study is good study expressing the relationship between sleep quality and restless leg syndrome.

In results section, authors have mentioned about the percentage of anxiety and depression symptoms but please mention why there was necessity to check these symptoms with sleep quality

Please resolve typing and grammar errors (example in line number 92, 95)

Reviewer #2: this neurological condition is novel as very few people talk about this. also it involves approx. 3000 students. so the research is good. I have 3 submissions:

1. The author has not provided the questionnaire of google form given to the students. many unreported/Asyptomatic cases may have been used in this study.

2. The authors should provide adequate justification for the use of the software in this study. They should explain why this particular software was chosen and how it aligns with the research objectives. By elaborating on the software's specific functionalities and how it supports the analysis of the data, the authors can enhance the validity and reliability of their findings. also the detailed statistics must be checked from a experienced biostatician.

3. Control group is missing.

Reviewer #3: Copaja-Corzo et al conducted a study to evaluate the association between restless legs syndrome and sleep quality in Peruvian medical students, using cross-sectional survey data. And I would like to provide my feedback on the manuscript and recommend revisions for its improvement.

1. Inconsistency: several inconsistent descriptions appeared, and I would highly suggest the authors to examine carefully throughout. Examples are as follows:

* In the abstract: “22.8% had poor sleep quality” was used, however, in the main body and the result table, the correct number should be “77.3%”

* PR (Prevalence ratio) was misused with “RP”

* The data sample was either “3139” (used in main body) or “3129” (used in the abstract)?

2. Data collection: in the discussion, it was mentioned that the data was collected during the pandemic. It’s suggested to elaborate more on this in the Methods section.

3. Discussion: “Although an association was found in our study, the difference between those who present RLS and those who do not is very small (5%).” This conclusion may not be applicable, since the reference study used odds vs. prevalence ratio was used in this study.

Reviewer #4: Dear Editor,

I have reviewed the manuscript entitled "Association between restless legs syndrome and sleep quality in Peruvian medical students." Overall, the article is well written and the methodology is convincing. However, there are a few areas where additional clarification and detail would enhance the manuscript's clarity and robustness.

Firstly, in the Methods section, the authors should explain more precisely why they decided to use the Pittsburgh Sleep Quality Index (PSQI). It would be beneficial to discuss the strengths and weaknesses of the PSQI, with relative references, to justify its selection over other sleep quality assessment tools. This context would help readers understand the appropriateness of the PSQI for this particular study population.

Similarly, the choice to use the Hopkins Symptom Checklist-25 (HSCL-25) to assess anxiety and depression symptoms requires further elaboration. The authors should provide a detailed rationale for this selection, including a discussion of the tool's strengths and weaknesses, supported by relevant references. This additional information would strengthen the methodological justification and provide a clearer picture of the assessment process.

Moreover, in the Discussion section, the single variables considered for the association between restless legs syndrome and sleep quality in Peruvian medical students should be explained more precisely. In particular, the authors should discuss the potential influence of university location and nomophobia on the observed associations. This detailed analysis would offer deeper insights into the factors affecting the relationship between restless legs syndrome and sleep quality, thereby enhancing the discussion's comprehensiveness.

In conclusion, while the article is well written and methodologically sound, these suggested improvements would provide additional clarity and depth, thereby strengthening the manuscript further.

Reviewer #5: Table 1, for the variable type of university, shows that the category with the highest prevalence is the private university with 50.8%. This differs from what is quoted in the item RESULTS (line spacing 3 and 4), where it is literally stated that “approximately half of them (50.8%) studied at a public university”. It is crucial to correct this interpretation to ensure an accurate understanding of the findings.

In the RESULTS section (lines 6 and 7), it is noted that the point estimate of poor sleep quality (22.8%) is not within the 95% confidence interval [25.9%-29.1%]. Furthermore, in Table 1, for the variable Poor Sleep Quality (PSQI), a prevalence of 77.3% is reported, which differs significantly from the 22.8% indicated both in the summary and in the results. It is recommended that the information be verified and corrected to ensure the consistency and accuracy of the data presented.

In Table 1, variable “year of studies”, it is recommended that the percentages of the categories (first, second, third, fourth, fifth) be revised and adjusted so that the sum is exactly 100.0%.

In Table 1, variable “sleep quality (PSQI)”, it is recommended that the percentages of the categories (good, bad) be revised and adjusted so that the sum is exactly 100.0%.

In the interpretation of table 2 (line 5), there is a typographical error in the notation of the percentage, indicated literally as “engaged in physical activity (72.4.9%). It is recommended that it be revised as appropriate to the data being reported.

It is recommended that in the interpretation of Table 3, not only the condition of risk factor but also the protective factor that is evident for the variables physical activity where the PAR was 0.95 [0.91 - 0.99] p=0.020 from which it can be deduced that medical students who perform physical activities presented 5.0% less probability of presenting poor sleep quality, similar condition to interpret for fifth year students PAR: 0.91 IC95% (0.84 - 0.98) p=0.013.

Reference to calculate the percentage of probability of protection used for the variable “physical activity”:

0.95-1=0.05 x 100=5.0% less probability of presenting poor sleep quality.

5.0% (RPA=0.95, IC 95%: 0.91 - 0.99)

Reviewer #6: The manuscript is clearly written and focused.I am recommending this manuscript for acceptance.

You need to expand your introduction/background with more information. Authors advised to recheck the laguage.

Reviewer #7: It is an interesting topic that shows the relationship between the PSQI and IRLSSG variables and it is important to improve some points:

Summary: it does not specify the type and design of study.

Methodology: it does not detail the research design and does not place the reliability value of the IRLSSG, the sequence of data collection is missing.

Discussion: The discussion needs to be improved; among the limitations, it should consider that the results could have been affected by some intervening variables, and a follow-up study could be carried out in this post-pandemic period.

Reviewer #8: I welcomed the opportunity to review this manuscript. The manuscript is very brief, but fascinating;

and I could read it within my scope. This study evaluated the association between RLS and sleep

quality in Peruvian medical students. Findings showed that students with RLS symptoms had a 5%

higher prevalence of poor sleep quality than those without RLS. Moreover, the prevalence of poor

sleep quality was very high (77.3%). It must be said that this study has considerable academic value

and practical significance.

However, I believe this manuscript leaves room for improvement, which means that the manuscript

needs a certain degree of revision (if we must define the degree, it can be said to be a major

revision). I will explain my opinions one by one below and request the author to respond to the

review comments one by one and revise them careful.

Please see my review as an attachment.

Reviewer #9: Dear author,

In this study, you investigated the relationship between restless legs syndrome and sleep quality among medical students. Although this topic has been investigated before, the high number of participants in your study is noteworthy. The manuscript is written in great detail. However, it needs minor revisions, which I have outlined below.

- The language of the manuscript should be revised.

- Spelling mistakes should be corrected.

- Page 13, Line 139-40. The information here does not match the table.

Best wishes

Reviewer #10: Dear authors: Epidemiological data are always required in the introduction. Furthermore, a non-probabilistic sample was chosen, which does not guarantee that these results are valid for the general population and, finally, there is no biological plausibility to explain the relationship between both variables.

6. PLOS authors have the option to publish the peer review history of their article (what does this mean? ). If published, this will include your full peer review and any attached files.

**Do you want your identity to be public for this peer review?** For information about this choice, including consent withdrawal, please see our Privacy Policy .

Reviewer #1: **Yes: ** Amit Kumar

Reviewer #2: No

Reviewer #3: No

Reviewer #4: No

Reviewer #5: **Yes: ** Huamani-Echaccaya Jose

Reviewer #6: **Yes: ** Dr Ravinder Singh

Reviewer #7: No

Reviewer #8: No

Reviewer #9: No

Reviewer #10: **Yes: ** J. Pierre Zila-Velasque

---

## [Author Response · Author response to Decision Letter 0]

23 Sep 2024

Dear editor,

Many thanks for your response to the submission of our paper

We have thoroughly revised the paper in response to the reviewers' comments. Our detailed responses to each of their points are provided below, with the original comments presented in italics.

Journal requirements

Reply: We have reviewed and modified the manuscript to ensure it meets the style criteria.

2. Please ensure that you have specified a) Did participants provide their written or verbal informed consent to participate in this study?

Reply: We have added that informed consent was obtained to participate in the questionnaire. The following was added: "In order for the students to participate in the survey, they first had to provide their informed consent."

Reply: We have added the study's database as supplementary material (Supplementary material 2).

Reviewer #1.

Overall, this study is good study expressing the relationship between sleep quality and restless leg syndrome.

In results section, authors have mentioned about the percentage of anxiety and depression symptoms but please mention why there was necessity to check these symptoms with sleep quality

Please resolve typing and grammar errors (example in line number 92, 95)

Reply: As suggested, we have corrected the grammatical errors in those lines.

Reviewer #2.

This neurological condition is novel as very few people talk about this. also it involves approx. 3000 students. so the research is good. I have 3 submissions:

1. The author has not provided the questionnaire of google form given to the students. many unreported/Asyptomatic cases may have been used in this study.

Reply: We have added the questionnaire as supplementary material (Supplementary material 1).

2. The authors should provide adequate justification for the use of the software in this study. They should explain why this particular software was chosen and how it aligns with the research objectives. By elaborating on the software's specific functionalities and how it supports the analysis of the data, the authors can enhance the validity and reliability of their findings. also the detailed statistics must be checked from a experienced biostatician.

Reply: We recognize the validity of the observation. , but STATA was created in 1985, is frequently updated and is a well-established software widely recognized in health, economic, and social sciences, and supports conducting adjusted regressions. The analysis of this study was carried out by a psychologist trained in statistics.

3. Control group is missing.

Reply: We conducted a cross-sectional analytical study, in which the independent variable was restless legs syndrome (RLS), and the control group consisted of participants without RLS. Table 1 presents the total number of individuals with and without RLS, and Table 3 identifies the “reference group” (i.e., control group) as individuals without RLS.

Reviewer #3.

Copaja-Corzo et al conducted a study to evaluate the association between restless legs syndrome and sleep quality in Peruvian medical students, using cross-sectional survey data. And I would like to provide my feedback on the manuscript and recommend revisions for its improvement.

1. Inconsistency: several inconsistent descriptions appeared, and I would highly suggest the authors to examine carefully throughout. Examples are as follows:

* In the abstract: “22.8% had poor sleep quality” was used, however, in the main body and the result table, the correct number should be “77.3%”

Reply: We agree. In the abstract, we have changed “22.8% had poor sleep quality” to “77.3% had poor sleep quality”. In the Results section, we have changed “Additionally, 15.3% presented RLS symptoms, while 22.8% (95% CI: 25.9% - 29.1%) had poor sleep quality” to “Additionally, 15.3% presented RLS symptoms, while 77.3% had poor sleep quality”.

* PR (Prevalence ratio) was misused with “RP”

Reply: We have corrected the grammatical error.

* The data sample was either “3139” (used in main body) or “3129” (used in the abstract)?

Reply: In the abstract, we have changed “3129” to “3139”.

2. Data collection: in the discussion, it was mentioned that the data was collected during the pandemic. It’s suggested to elaborate more on this in the Methods section.

Reply: We have added more details about the data collection in the Methods section:

“The primary study collected data in two stages between June 2020 and March 2021. This was done virtually through social media platforms (Facebook, WhatsApp) because universities in Peru held online classes during the pandemic to minimize the risk of contagion.

The data collection occurred in two stages. In the first stage, a Facebook page was created to promote the research project. An open invitation was extended to all medical students in Peru through posters distributed in private Facebook groups for medical students, as well as through paid Facebook ads specifically targeting medical students in Peru.

In the second stage, 23 students from various medical schools were recruited through networks within the Scientific Society of Medical Students of Peru. The recruited students attended a 45-minute virtual meeting, where they were trained to contact students from their respective universities. As an incentive, all students who completed the survey were given free access to a Google Drive folder with a collection of resources from medical courses”.

3. Discussion: “Although an association was found in our study, the difference between those who present RLS and those who do not is very small (5%).” This conclusion may not be applicable, since the reference study used odds vs. prevalence ratio was used in this study.

Reply: We changed the sentence and specified that we are referring to the prevalence in Discussion section: “although an association was found in our study, the difference in prevalence between those who present RLS and those who do not is very small (5%)”.

Reviewer #4.

Dear Editor,

I have reviewed the manuscript entitled "Association between restless legs syndrome and sleep quality in Peruvian medical students." Overall, the article is well written and the methodology is convincing. However, there are a few areas where additional clarification and detail would enhance the manuscript's clarity and robustness.

Firstly, in the Methods section, the authors should explain more precisely why they decided to use the Pittsburgh Sleep Quality Index (PSQI). It would be beneficial to discuss the strengths and weaknesses of the PSQI, with relative references, to justify its selection over other sleep quality assessment tools. This context would help readers understand the appropriateness of the PSQI for this particular study population.

Reply: According to a review that compares and discusses sleep quality questionnaires, we have added in the Methods section: “The PSQI is the most widely used measure for assessing sleep quality, useful for epidemiological studies and easy to complete. A weakness is that different factorial structures have been found in the samples where it has been applied.”

Similarly, the choice to use the Hopkins Symptom Checklist-25 (HSCL-25) to assess anxiety and depression symptoms requires further elaboration. The authors should provide a detailed rationale for this selection, including a discussion of the tool's strengths and weaknesses, supported by relevant references. This additional information would strengthen the methodological justification and provide a clearer picture of the assessment process.

Reply: According to a study that evaluate the psychometric properties of the HSCL-25 in a Spanish version, we have added in Method section: “HSCL-25 is widely used for both clinical and epidemiological, self administered, easy-to-use for participants and researchers and simple of interpretation. It does not provide a clinical diagnosis; it only assesses early symptoms of anxiety and depression”.

Moreover, in the Discussion section, the single variables considered for the association between restless legs syndrome and sleep quality in Peruvian medical students should be explained more precisely. In particular, the authors should discuss the potential influence of university location and nomophobia on the observed associations. This detailed analysis would offer deeper insights into the factors affecting the relationship between restless legs syndrome and sleep quality, thereby enhancing the discussion's comprehensiveness.

Reply: Thank you for your commentary. In the analysis we carried out we have adjusted for both university location and nomophobia. We adjusted the statistical analysis by location since we thought it was a variable that could influence. In this way, we found that the association between restless legs syndrome was presented in the crude and multivariate analysis

In conclusion, while the article is well written and methodologically sound, these suggested improvements would provide additional clarity and depth, thereby strengthening the manuscript further.

Reviewer #5.

Table 1, for the variable type of university, shows that the category with the highest prevalence is the private university with 50.8%. This differs from what is quoted in the item RESULTS (line spacing 3 and 4), where it is literally stated that “approximately half of them (50.8%) studied at a public university”. It is crucial to correct this interpretation to ensure an accurate understanding of the findings.

Reply: Thanks for the comment. We have corrected the sentence.

In the RESULTS section (lines 6 and 7), it is noted that the point estimate of poor sleep quality (22.8%) is not within the 95% confidence interval [25.9%-29.1%]. Furthermore, in Table 1, for the variable Poor Sleep Quality (PSQI), a prevalence of 77.3% is reported, which differs significantly from the 22.8% indicated both in the summary and in the results. It is recommended that the information be verified and corrected to ensure the consistency and accuracy of the data presented.

Reply: As suggested (also by reviewer 3), we have verified and corrected the sentences in the Results section.

In Table 1, variable “year of studies”, it is recommended that the percentages of the categories (first, second, third, fourth, fifth) be revised and adjusted so that the sum is exactly 100.0%.

Reply: We agree. We have modified and the sum is now exact.

In Table 1, variable “sleep quality (PSQI)”, it is recommended that the percentages of the categories (good, bad) be revised and adjusted so that the sum is exactly 100.0%.

Reply: We have changed “77.3%” to “77.2%” in all the manuscript.

In the interpretation of table 2 (line 5), there is a typographical error in the notation of the percentage, indicated literally as “engaged in physical activity (72.4.9%). It is recommended that it be revised as appropriate to the data being reported.

Reply: It was actually a typographical error. We have corrected it.

It is recommended that in the interpretation of Table 3, not only the condition of risk factor but also the protective factor that is evident for the variables physical activity where the PAR was 0.95 [0.91 - 0.99] p=0.020 from which it can be deduced that medical students who perform physical activities presented 5.0% less probability of presenting poor sleep quality, similar condition to interpret for fifth year students PAR: 0.91 IC95% (0.84 - 0.98) p=0.013.

Reference to calculate the percentage of probability of protection used for the variable “physical activity”:

0.95-1=0.05 x 100=5.0% less probability of presenting poor sleep quality.

5.0% (RPA=0.95, IC 95%: 0.91 - 0.99).

Reply: In this study, we aim to assess the association between RLS and sleep quality. To better understand this relationship, we are considering potential confounding variables, including physical activity, that could influence the association. Therefore, the adjusted PR for physical activity in Table 3 is not directly interpretable.

Reviewer #6.

The manuscript is clearly written and focused.I am recommending this manuscript for acceptance.

You need to expand your introduction/background with more information. Authors advised to recheck the language.

Reply: We have added more information about epidemiological data and have better explained the research hypothesis, which can be seen in lines 57 to 60 and 66 to 75 of the introduction. Additionally, we have thoroughly reviewed the language.

Reviewer #7.

It is an interesting topic that shows the relationship between the PSQI and IRLSSG variables and it is important to improve some points:

Summary: it does not specify the type and design of study.

Reply: We have added in the abstract: “Cross-sectional study with secondary data analysis”.

Methodology: it does not detail the research design and does not place the reliability value of the IRLSSG, the sequence of data collection is missing.

Reply: We have specified that this is a cross-sectional study with a secondary data analysis. As suggested (also by reviewer 3), we have added information about data collection in the Methods section. Regarding the measure for restless legs syndrome, although this instrument has not undergone formal psychometric evaluation, we rely on a study that reported a sensitivity of 100% and a specificity of 88%. We have included: “This measure has a sensitivity of 100% and specificity of 88% in a Spanish version”.

Discussion: The discussion needs to be improved; among the limitations, it should consider that the results could have been affected by some intervening variables, and a follow-up study could be carried out in this post-pandemic period.

Reply: We have included in discussion section: “Fourth, other variables that could infuence in the association of interest, such as stress and the use of psychoactive substance, were not included”.

Reviewer #8.

I welcomed the opportunity to review this manuscript. The manuscript is very brief, but fascinating; and I could read it within my scope. This study evaluated the association between RLS and sleep quality in Peruvian medical students. Findings showed that students with RLS symptoms had a 5% higher prevalence of poor sleep quality than those without RLS. Moreover, the prevalence of poor sleep quality was very high (77.3%). It must be said that this study has considerable academic value and practical significance.

However, I believe this manuscript leaves room for improvement, which means that the manuscript needs a certain degree of revision (if we must define the degree, it can be said to be a major revision). I will explain my opinions one by one below and request the author to respond to the review comments one by one and revise them careful.

Please see my review as an attachment.

Reply: Thanks for your comment. We believe that your comments have helped improve the quality of the final manuscript.

1. Title

The title clearly states the core— restless legs syndrom

---

## [Decision Letter · Decision Letter 1]

20 Nov 2024

PONE-D-24-08629R1Association between restless legs syndrome and sleep quality in Peruvian medical studentsPLOS ONE

Dear Dr. Copaja-Corzo,

Thank you for submitting your manuscript to PLOS ONE. After careful consideration, we feel that it has merit but does not fully meet PLOS ONE’s publication criteria as it currently stands. Therefore, we invite you to submit a revised version of the manuscript that addresses the points raised during the review process.

We look forward to receiving your revised manuscript.

Kind regards,

Runtang Meng, PhD

Academic Editor

PLOS ONE

Journal Requirements:

Additional Editor Comments:

1. It is recommended that the model fitting methodology be based on literature references that clearly state that the adjustment or confounding variables are not directly interpretable, allowing the interpretation to focus exclusively on the independent variable or variable of interest.

I suggest that once the relevant references have been cited, a clear specification of the independent variable, the dependent variable and the adjustment or confounding variables should be included in the methodology section. In addition, it would be advisable to add a footnote to the relevant table indicating that certain variables are not interpretable and giving a brief justification. This will provide greater clarity for the reader and reduce the risk of misinterpretation of the role of the adjustment variables.

You may find it useful to consult the following reference:

Akinkugbe AA, Simon AM, Brody ER. A scoping review of the Table 2 fallacy in the oral health literature. Community Dent Oral Epidemiol. 2021 Apr;49(2):103-109. doi: 10.1111/cdoe.12617.

2. A Poisson regression model with robust variance was used in the multivariate analysis. It is recommended that the authors justify the choice of this model instead of logistic regression, as this explanation provides important information to the reader. The following reference may be helpful: Thompson ML, Myers JE, Kriebel D. Prevalence odds ratio or prevalence ratio in the analysis of cross-sectional data: what to do? Occup Environ Med. 1998 Apr;55(4):272-7. doi: 10.1136/oem.55.4.272.

Reviewers' comments:

Reviewer's Responses to Questions

**Comments to the Author**

1. If the authors have adequately addressed your comments raised in a previous round of review and you feel that this manuscript is now acceptable for publication, you may indicate that here to bypass the “Comments to the Author” section, enter your conflict of interest statement in the “Confidential to Editor” section, and submit your "Accept" recommendation.

Reviewer #1: All comments have been addressed

Reviewer #5: All comments have been addressed

Reviewer #8: All comments have been addressed

Reviewer #9: All comments have been addressed

2. Is the manuscript technically sound, and do the data support the conclusions?

Reviewer #1: Yes

Reviewer #5: Yes

Reviewer #8: Yes

Reviewer #9: Yes

3. Has the statistical analysis been performed appropriately and rigorously? 

Reviewer #1: Yes

Reviewer #5: Yes

Reviewer #8: Yes

Reviewer #9: Yes

4. Have the authors made all data underlying the findings in their manuscript fully available?

Reviewer #1: Yes

Reviewer #5: Yes

Reviewer #8: Yes

Reviewer #9: Yes

5. Is the manuscript presented in an intelligible fashion and written in standard English?

Reviewer #1: Yes

Reviewer #5: Yes

Reviewer #8: Yes

Reviewer #9: Yes

6. Review Comments to the Author

Reviewer #1: (No Response)

Reviewer #5: 1. It is recommended that the model fitting methodology be based on literature references that clearly state that the adjustment or confounding variables are not directly interpretable, allowing the interpretation to focus exclusively on the independent variable or variable of interest.

I suggest that once the relevant references have been cited, a clear specification of the independent variable, the dependent variable and the adjustment or confounding variables should be included in the methodology section. In addition, it would be advisable to add a footnote to the relevant table indicating that certain variables are not interpretable and giving a brief justification. This will provide greater clarity for the reader and reduce the risk of misinterpretation of the role of the adjustment variables.

You may find it useful to consult the following reference:

Akinkugbe AA, Simon AM, Brody ER. A scoping review of the Table 2 fallacy in the oral health literature. Community Dent Oral Epidemiol. 2021 Apr;49(2):103-109. doi: 10.1111/cdoe.12617.

2. A Poisson regression model with robust variance was used in the multivariate analysis. It is recommended that the authors justify the choice of this model instead of logistic regression, as this explanation provides important information to the reader. The following reference may be helpful: Thompson ML, Myers JE, Kriebel D. Prevalence odds ratio or prevalence ratio in the analysis of cross-sectional data: what to do? Occup Environ Med. 1998 Apr;55(4):272-7. doi: 10.1136/oem.55.4.272.

Reviewer #8: The author has satisfactorily addressed all my comments, demonstrating a thorough understanding of the feedback provided. The revisions have significantly improved the manuscript's clarity and rigor. I am confident in the quality of the work and therefore recommend it for publication in the journal.

Reviewer #9: (No Response)

7. PLOS authors have the option to publish the peer review history of their article (what does this mean? ). If published, this will include your full peer review and any attached files.

**Do you want your identity to be public for this peer review?** For information about this choice, including consent withdrawal, please see our Privacy Policy .

Reviewer #1: No

Reviewer #5: **Yes: ** Huamani-Echaccaya Jose Luis

Reviewer #8: No

Reviewer #9: No

---

## [Author Response · Author response to Decision Letter 1]

9 Dec 2024

Dear editor,

Many thanks for your response to the submission of our paper

Our detailed responses to each of their points are provided below, with the original comments presented in italics.

1. It is recommended that the model fitting methodology be based on literature references that clearly state that the adjustment or confounding variables are not directly interpretable, allowing the interpretation to focus exclusively on the independent variable or variable of interest.

I suggest that once the relevant references have been cited, a clear specification of the independent variable, the dependent variable and the adjustment or confounding variables should be included in the methodology section. In addition, it would be advisable to add a footnote to the relevant table indicating that certain variables are not interpretable and giving a brief justification. This will provide greater clarity for the reader and reduce the risk of misinterpretation of the role of the adjustment variables.

You may find it useful to consult the following reference:

Akinkugbe AA, Simon AM, Brody ER. A scoping review of the Table 2 fallacy in the oral health literature. Community Dent Oral Epidemiol. 2021 Apr;49(2):103-109. doi: 10.1111/cdoe.12617.

- Reply: We have added the following in Data analysis section:

“It is essential to emphasize that only the estimate of the association between RLS and sleep quality should be interpreted, while the estimates for the confounding variables should not be considered30”.

We also included in Table 3:

“**Only this adjusted RP should be interpreted, the rest of adjusted RP correspond to the confounding variables so they should not be interpreted”.

Finally, we included in Variables section which was the independent, dependent and confounding variables.

2. A Poisson regression model with robust variance was used in the multivariate analysis. It is recommended that the authors justify the choice of this model instead of logistic regression, as this explanation provides important information to the reader. The following reference may be helpful: Thompson ML, Myers JE, Kriebel D. Prevalence odds ratio or prevalence ratio in the analysis of cross-sectional data: what to do? Occup Environ Med. 1998 Apr;55(4):272-7. doi: 10.1136/oem.55.4.272

- Reply: We used this regression model since the frequency of inadequate sleep quality was high so using logistic regression (i.e., calculating odds ratios) could overestimate the objective association29

---

## [Decision Letter · Decision Letter 2]

12 Feb 2025

Association between restless legs syndrome and sleep quality in Peruvian medical students

PONE-D-24-08629R2

Dear Dr. Copaja-Corzo,

We’re pleased to inform you that your manuscript has been judged scientifically suitable for publication and will be formally accepted for publication once it meets all outstanding technical requirements.

Kind regards,

Runtang Meng, PhD

Academic Editor

PLOS ONE

Additional Editor Comments (optional):

Reviewers' comments:

Reviewer's Responses to Questions

**Comments to the Author**

1. If the authors have adequately addressed your comments raised in a previous round of review and you feel that this manuscript is now acceptable for publication, you may indicate that here to bypass the “Comments to the Author” section, enter your conflict of interest statement in the “Confidential to Editor” section, and submit your "Accept" recommendation.

Reviewer #5: All comments have been addressed

Reviewer #8: All comments have been addressed

Reviewer #9: All comments have been addressed

2. Is the manuscript technically sound, and do the data support the conclusions?

Reviewer #5: Yes

Reviewer #8: Yes

Reviewer #9: Yes

3. Has the statistical analysis been performed appropriately and rigorously? 

Reviewer #5: Yes

Reviewer #8: Yes

Reviewer #9: Yes

4. Have the authors made all data underlying the findings in their manuscript fully available?

Reviewer #5: Yes

Reviewer #8: Yes

Reviewer #9: (No Response)

5. Is the manuscript presented in an intelligible fashion and written in standard English?

Reviewer #5: Yes

Reviewer #8: Yes

Reviewer #9: Yes

6. Review Comments to the Author

Reviewer #5: (No Response)

Reviewer #8: Based on my review, the manuscript meets the journal's standards and requirements. The authors have adequately addressed all comments and concerns raised during the review process. I find the study to be scientifically sound, well-structured, and ready for publication in its current form.

Reviewer #9: (No Response)

7. PLOS authors have the option to publish the peer review history of their article (what does this mean? ). If published, this will include your full peer review and any attached files.

**Do you want your identity to be public for this peer review?** For information about this choice, including consent withdrawal, please see our Privacy Policy .

Reviewer #5: **Yes: ** Huamani-Echaccaya Jose Luis

Reviewer #8: No

Reviewer #9: No

---

## [Editor Report · Acceptance letter]

PONE-D-24-08629R2

PLOS ONE

Dear Dr. Copaja-Corzo,

I'm pleased to inform you that your manuscript has been deemed suitable for publication in PLOS ONE. Congratulations! Your manuscript is now being handed over to our production team.

Kind regards,

on behalf of

Dr. Runtang Meng

Academic Editor

PLOS ONE